# Traceability Platform Based on Green Blockchain: An Application Case Study in Dairy Supply Chain

**Giuseppe Varavallo** [1,*] **, Giuseppe Caragnano** [1] **, Fabrizio Bertone** [1] **, Luca Vernetti-Prot** [2] **and Olivier Terzo** [1]

1   LINKS Foundation, via Boggio 61, 10138 Turin, Italy; giuseppe.caragnano@linksfoundation.com (G.C.);
    fabrizio.bertone@linksfoundation.com (F.B.); olivier.terzo@linksfoundation.com (O.T.)
2   Institut Agricole Régional (IAR), Regione la Rochere 1, 11100 Aosta, Italy; l.vernetti@iaraosta.it (L.V.-P.)
*   Correspondence: giuseppe.varavallo@linksfoundation.com

**Abstract:** Recent progress in IoT and software development has simplified data acquisition and immutability of information in the agri-food supply chain. In the last few years, several frameworks and applications were proposed to ensure traceability in the agri-food-sector using distributed ledger technologies (DLT) such as Blockchain technologies. Still, no other study has presented a Blockchain-based traceability platform with a lower impact on the environment and lower cost for each transaction sent by the supply chain. This article presents a traceability platform based on Green Blockchain with low energy consumption and costs savings applied to the Fontina PDO cheese supply chain, part of the project "Typicalp", funded by the European Union (EU). The proposed traceability system is based on Algorand Blockchain, which uses the Pure Proof-of-Stake mechanism of consensus that requires minimal computational power, is highly scalable and environmentally sustainable. In addition to the environmental and financial benefits, the developed traceability platform has made it possible to digitize the entire production chain, making the data immutable and available in real-time for Fontina consortium operators and final consumers.

**Keywords:** traceability platform; green blockchain; sustainability; dairy industry; supply chain

## 1. Introduction

The dairy supply chain is among the most complex of the food supply chain, as it comprises a multiplicity of actors and transformation processes. It is the second most relevant agricultural sector in the EU [1], accounting for over 12% of total agricultural production. More than a million producers annually put on the market millions of tons of cheese, butter, and milk. According to the ISMEA [2] (Institute of Services for the Agricultural Food Market), the dairy supply chain's main players are feed companies, farmers, milk collection centers, transporters, and milk cooperatives. Italy stands out in the dairy sector with more than 50 PDO (Protected Designation of Origin) cheeses, 2 PGI (Protected Geographical Indication) and 1 TSG (Traditional Specialty Guaranteed, mozzarella). Regarding domestic consumption of milk and dairy products, the percentage share in the value of domestic consumption is 60% for cheeses and dairy products, 13% for UHT milk, 12% for yogurt, and 7% for fresh milk, 3% for butter, and 2% for cream (2018). Food traceability is crucial for food quality and safety and is one of the most significant challenges to assuring sustainability in food supply chains. The globalization of the food supply chain increases food product vulnerabilities. Therefore, traceability is fundamental for managing recalls in supply chains and minimizing risks [3].

More specifically, traceability is the ability to reproduce the history of products and follow the use of each raw material through all stages of production, transformation, and distribution [4]. This is done through documented identification of material flows and supply chain operators. The implementation of a traceability system, which offers the possibility of being able to trace the origin of the materials used for a food product, is essential for the following aspects [5]:

- guarantee of safety during food production;
- identification of the causes of the problems;
- attribution of responsibilities between supply chain operators;
- improvement of the quality of the products supplied.

Recent progress in IoT and software development [6,7] has simplified data acquisition in the production chain. This is particularly useful to collect and monitor critical parameters such as temperature, humidity, and timestamp of events. To ensure the immutability of data and therefore guarantee compliance with regulations, many platforms, especially in the agri-food sector, are taking advantage of the Distributed Ledger Technologies (DLT) such as Blockchains [8,9]. Several authors have identified the benefits of applying DLT in the agri-food sector and in particular in the dairy sector [10–12]. The principal benefits are for producers that can more efficiently communicate with the other participants of the dairy supply chain system. In addition, it will enhance the confidence of consumers in the quality of food they purchase [11,13]. The usage of DLT in the logistics supply chain will ensure the reliability of the data, remove the possibility of changing information about the production and transportation phases, and establish trust, efficiency, quality, and resilience [14–16]. The Blockchain is a shared (distributed) and decentralized ledger. This is probably the most significant idea behind the Blockchain, which differs from databases or regular ledgers. On a shared ledger, the data is not in one place, but there are thousands of secure copies of that data on different computers (nodes) worldwide. This increases data security as it is much easier to hack into a centralized ledger with only one copy of the data than thousands of copies in a shared ledger. Bitcoin is the first application of "Blockchain technology", which relies on cryptographic algorithms and peer-to-peer technologies [17]. Revealed by Satoshi Nakamoto in 2008, Blockchain technology is not only at the foundation of all cryptocurrencies, but it also unlocks the door to new applications such as smart contracts [18].

However, in the last few years, Distributed Ledger technologies have started attracting negative attention as they require high energy consumption due to the Proof-of-Work (PoW) [19] algorithms commonly used to guarantee consensus within the distributed network, drawing stern criticism from academia, business, and the media [20]. According to Schinckus [21], the electricity demanded to support the bitcoin market is higher than the annual consumption of some advanced countries such as Switzerland or Denmark.

To mitigate this critical issue, new energy-efficient consensus algorithms are being proposed and implemented, such as Proof of Stake (PoS) [22] algorithms.

This study concerns the traceability of Fontina PDO cheese (https://www.fontina-dop.it/ (accessed on 18 Febuary 2022)) produced in the Aosta Valley region (Italy), particularly the supply chain formed by different operators/entities: farmers, dairies, transporters, seasoning operators, distributors, and the Fontina Consortium.

The main objective of the work is to illustrate a traceability platform based on Green Blockchain with low energy consumption, minimum environmental impact, and cost savings applied to the Fontina PDO supply chain. It was chosen to use the Algorand blockchain [23] as it applies a Pure Proof-of-Stake mechanism of validation, which is a new consensus procedure that requires minimal computational power and is highly scalable. Furthermore, the Fontina Consortium aims to promote and guarantee the data collected on the Fontina PDO supply chain, which was proposed as part of the project "Typicalp (https://www.progetti.interreg-italiasvizzera.eu/it/b/78/typicalp (accessed on 18 Febuary 2022))", funded by the European Union (EU). The rest of the article is structured as follows: the Section 2 provides a literature review regarding the Distributed Ledger Technologies and traceability platforms employed in the Dairy industry. Particularly, different Consensus Mechanism is considered with their strengths and limits. Section 3 presents the research methods employed to analyze the Fontina PDO cheese supply chain and current traceability system. Section 4 offers a detailed description of the proposed Traceability platform based on Green Blockchain. In this section are specified: Back-end framework; Relational Database Management System (RDBMS); Distributed ledger

technology (DLT) applied to the Fontina PDO cheese supply chain. Finally, the paper ends with a discussion of results and a conclusion section.

## 2. Literature Review

### 2.1. Distributed Ledger Technologies

Distributed ledger Technologies (DLT) are a broad "family" of technologies that generate distributed archives on the network [24]. A *ledger* is a tool used to record transactions. The Blockchain is a digital ledger structured as a chain of blocks ("registers") responsible for data collection. The distributed archive was born with the "genesis" block where the first information is stored. It is possible to insert other blocks of information, but it is not possible to modify and remove blocks previously added to the chain [25]. The security and immutability of information within the blocks are guaranteed by a consensus protocol and encryption.

Figure 1 shows the various steps that comprise a Blockchain network. The various phases to record transactions in Blockchain technology is described below:

1. Someone requests a transaction;
2. The transaction is represented online as block;
3. The requested transaction is sent to a peer to peer network made up of computers known as Nodes;
4. The network of nodes validates the transaction and the user's identity using consensus algorithms. A transaction may involve cryptocurrency, contracts, records or others information;
5. Next, the new block is added to the existing blockchain in a way that it can be permanent and unalterable;
6. The transaction is complete.

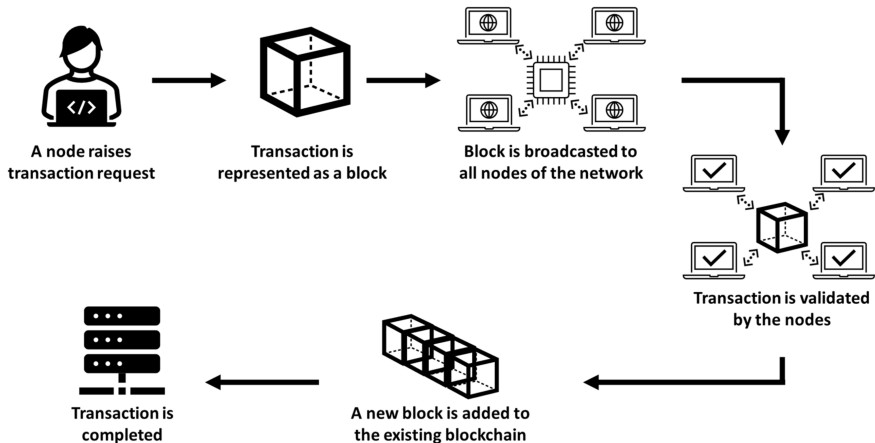

**Figure 1.** Steps to record transactions in a Blockchain network. (Reprinted from [26]).

### 2.2. Consensus Mechanisms: Analysis of Different Protocol

In the context of the Blockchain, consensus means that two or more parties agree on the correct state of the data on the system and synchronize the data on the Blockchain [27]. This means that each copy of the shared ledger will have the same data. Without consent, transactions lapse and are not saved in the Blockchain. Furthermore, the consensus mechanism serves to help two or more parties who do not trust each other, to trust each other [28]. Consensus mechanisms are the main factor that impacts the performance of a Blockchain and energy consumption [29]. There are several ways to reach a consensus. In particular, there are several consensus algorithms, the main ones are:

- Proof of Work (PoW) is a protocol used to reach distributed consensus in which voting power is based on computational power [30]. The PoW is based on hash algorithms. To add a block to the chain, the network nodes that are also known as ("PoW-mining")

compete to solve a complex mathematical problem, which usually requires a lot of computing power. To encourage the creation of new blocks, rewards are provided for the miners.

- Proof of Stake (PoS) is an alternative protocol employed in the process of reaching distributed consensus. The goal of the PoS is the same as the PoW, but the process to achieve the final goal is different [31]. Unlike the PoW where miners are rewarded by solving complex mathematical problems, in the PoS, there are validators (which can be equated to miners in the PoW) chosen in advance based on the amount of cryptocurrencies owned by the related Blockchain, defined as stake. In PoS-mining (staking), the owned tokens are used instead of the computing power possessed. Users in possession of tokens can "stake" their tokens (temporarily blocking the tokens until the staking process ends) to have the right to confirm the transaction of a block in exchange.

However, these two types of consensus mechanisms are recognized to be not environmentally sustainable. Nevertheless, the so-called third-generation Blockchains, such as Algorand, are trying to solve this problem and become fully carbon-neutral, with minimal environmental impact. Algorand deployed the Pure Proof-of-Stake consensus mechanism which is more environmentally sustainable and requires minimal electrical power.

*Pure Proof-of-Stake* is a variant of the proof of stake, in which the validators of a new block are not selected only based on the number of tokens they have staked but also randomly, through the Verifiable Random Function (VRF) [32]. The VRF behaves similarly to a weighted lottery; in practice, it is as if each participant who has ALGO coins in their wallet has a weighted quantity of lottery tickets to validate transactions. This validation mechanism has several advantages, in particular:

- *Speed*: the blocks are validated in a few seconds. Algorand transaction transferring is similar to that of large payment networks.
- *Advanced decentralization*: The pure proof of stake addresses the dilemma in an equidistant way by initially selecting validators at random and not just based on the number of tokens staked. Therefore, all blockchain users can be selected at some point to participate in the process of validation, which ensures security and a higher level of decentralization.
- *Low energy consumption*: The Verifiable Random Function does not require large amounts of energy or very complex hardware to run. In other words, a pure proof of stake validator does not have to invest considerable resources to participate in the network.

Algorand has implemented a new consensus procedure to solve many issues that influence the Proof-of-Work, and the Proof-of-Stake consensus mechanism. Therefore, the whole process is energy-saving and fully decentralized [33].

### 2.3. Traceability Platforms & Blockchain Technologies Employed in the Dairy Industry

The Blockchain is a technology capable of improving the transparency of the agri-food chain. This solution can create a flow of immutable data that any consumer can consult. Therefore, Blockchain Technologies need to be supported by a good traceability platform that guarantees a simple and constant flow of information.

Blockchain has the potential to revolutionize supply chain sustainability. Adopting Blockchain technology makes it possible to monitor production parameters such as energy consumption, raw materials, processing, and emissions. As a result, Blockchain technology can significantly decrease carbon emissions and air pollution [34]. Moreover, tracking substandard products can help reduce rework and decrease resource consumption [35]. It is also observed that increased visibility, automation, smart execution, and reliability parameters of Blockchain can help accentuate reusability, recycling, and circular performance management in the supply chain, leading to a high-quality food life cycle [36,37]. Not for the last one, the Blockchain can help to raise awareness about the environmental characteristics of the food produced. For example, raising awareness about land, soil, and

water degradation where food is produced. Furthermore, tracing this information via the supply chain makes it visible to the public [38].

In the last few years, numerous authors have proposed Blockchain-enabled traceability mechanisms [39,40]. Several authors have proposed traceability mechanisms and Blockchain applications for tracking agricultural products [41–43].

In the dairy supply chain, Regattieri et al. [44] have proposed a platform to support the traceability of the famous Italian cheese "Parmigiano Reggiano" from the farm to the final consumer. The framework supports identifying the product's characteristics in various aspects along the value chain: cattle breeding, dairy, seasoning, and packaging warehouse. The developed system is based on a central database that collects data at all identified stages in the food chain. Some information is collected automatically using sensors and barcodes, while other information is collected manually.

Manikas & Manos [45] have presented an integrated project for traceability dairy products by creating a web platform. The first phase of the project involves designing a reference model for a generic milk supply chain. The authors have identified three main phases in a dairy supply chain: natural environment (e.g., farm, raw material production), processing, and distribution. The authors also have identified the main entities involved: actors (e.g., breeder, dairy, seasoning workers, etc.), transporters, raw material, and the final sample (final batch). In the final part of their work, the authors presented the class diagram for each entity identified.

Magliulo et al. [46] created a platform for the traceability of dairy products through the "Bovlac" project. The project concerned the traceability of the Naples fiordilatte cheese through a system called ValueGo integrated with RFID labeling, which collects information at every stage of the supply chain. The information acquired in each production phase is shared in a portal that the consumer can consult by scanning the NFC tag on the package. The platform was also integrated with a farm surveillance system.

De las Morenas et al. [47] designed a prototype of a traceability system for the dairy industry, focusing mainly on the raw material, milk, a system with an RFID tag applied to the milk samples collected, and a GPS sensor. The milk samples collected are placed inside a portable container equipped with data acquisition and analysis sensors. The sensors were applied to track the samples' entire path, the time elapsed, and the temperature variations.

Vara Martinez et al. [48] , following the previous authors, also focused on the traceability of milk, building an integrated traceability system for different animals (cows, sheep, goats, etc.). In addition, in this system, thanks to RFID and GPS sensors' aid, it is possible to monitor the temperature, the path of the milk, and the time elapsed.

Giacalone et al. [49] showed in their research an innovative platform that connects the technology of the Internet of things and the Blockchain for the certification and protection of the production stages of the PDO food chain and specifically the Mozzarella PDO of Gioia del Colle. Furthermore, the developed modules are based on verifying the compliance of the geolocation and the possibility of tracing the batches of milk produced, destroyed, and delivered.

Rambim and Awuor [50] proposed Milk Delivery Blockchain Manager (MDBM). This decentralized platform automatically captures the quantity and quality of milk delivered by the farmer and the time and date when the milk is delivered. Blockchain enables the creation of records that are permanent and non-repudiated by all stakeholders, i.e., farmers and local milk collection centers.

Casino et al. [13] have developed fully functional smart contracts and a local private blockchain for Pagonis Sisters and Co., a dairy company based in Greece. The proposed architecture enables the tracking of products and processes. Therefore, it grants access to the full link between the goods provided to the consumers and the operations executed in the various supply chain processes.

Niya et al. [12] presented a Blockchain-based supply chain traceability system designed and implemented for the Swiss dairy supply chain. The developed application

facilitates automated data collection and employs Ethereum Blockchain based on the Proof of Stake mechanism.

Fang and Stone [51] have also proposed a new dairy logistics supply ecosystem with Blockchain technology. The data proposed in the system is open to all participants, and the Internet of things (IoT) lowers labor and reduces errors. The proposed dairy logistics supply system will use a local Blockchain instead of a public.

*2.4. Regulatory Overview*

The European Commission has issued many regulations relating to food safety and traceability in recent years. One of the most important mandatory standards is the EC Regulation 178/2002, which establishes the producers' obligation to ensure the traceability of food destined for humans, from the primary production stage to the marketing stage. In particular, art. 17 imposes the following obligations: "It is the responsibility of food and feed business operators to ensure that food or feed in the undertakings controlled by them complies with the provisions of food law relating to their activities at all stages of production, processing and verify that these provisions are met ". To meet these obligations and ensure food traceability, the actors in the supply chain must collect information in all its phases. This involves the application of Article 18 traceability: "Food and feed business operators must be able to identify who has supplied them with food, feed, food-producing animal or any substance intended or capable of entering to be part of a food or feed". Furthermore, the rules oblige all operators in the food chain to register and provide information relating to suppliers and direct customers. Transporters are also considered supply chain food operators and are obliged to record incoming and outgoing food products. However, the European regulation EC 178/2002 does not specify the methods for recording information. This poses several problems since each operator adopts a different traceability model that sometimes involves losing information in some phases of the supply chain. In several cases, especially for small producers, the information being recorded on paper is very difficult and time consuming to search in case of any issue discovered at a later stage, for example when the product is already on the market.

**3. Research Methods**

A requirements analysis was performed to design, develop, and implement the Blockchain-based traceability platform. The requirements were collected through different methodologies: document analysis, field research, interview & focus group.

*Document analysis:* A document analysis of the Fontina PDO cheese supply chain was carried out. The Fontina Consortium provides "Production Regulations [52]" that specify all steps and rules to produce Fontina PDO cheese.

*Field research*: Field research was fundamental to understanding the state of the art of the current traceability system of Fontina PDO cheese. The field research was performed in the diaries and mountain pasture in Aosta Valley.

*Interview & focus group*: The interview was conducted on 30 operators of the Fontina PDO production chain. Firstly, semistructured interviews were conducted for each operator with questions relating to two domains: User side, to understand how many operators interact within the Fontina PDO cheese production chain; Type of information, relates to the data to be tracked. Subsequently, a focus group was organized with all operators to collect information about the platform's financial and environmentally sustainable features.

The following paragraphs will describe the production phases of Fontina PDO cheese and the current traceability system in detail.

*3.1. Fontina PDO—Supply Chain Phases*

All the production phases have been described based on the document analysis and the "Production Regulations" provided by the Consorzio Fontina PDO.

Fontina is a semi-cooked full-fat cheese made with whole cow's milk, coming from a single milking. The processing, seasoning, and portioning of Fontina cheese occurs within

the Aosta Valley territory. The seasoning takes at least three months. Whole Fontina cheese is cylindrical (35–45 cm in diameter, 7–10 cm in height) and has a variable weight between 7.5 and 12 kg. Fontina PDO cheese is among the most important products in Italy. In the Aosta Valley region, 200 mountain pastures, 700 farms, and around 430,000 Fontina PDO cheese are produced yearly (2021) [53].

Figure 2 shows the production chain of Fontina PDO cheese. The supply chain stages of Fontina PDO cheese consist of five phases:

1.  *Milk production*: The primary raw material is cow's milk from mountain pastures in the Aosta Valley area. The milk collected for processing into Fontina cheese has the following required characteristics: raw milk, whole milk, coming from a single milking. Raw milk must be processed within two hours of milking cows. Each farmer delivers the single quantity of milk produced to the transporter in this phase.

2.  *Processing*: The transporter delivers the total milk collected to dairies for transformation. The milk delivered must not have undergone heating to a temperature above 36 °C. The milk collected is processed in cheese vat. The procedure must occur at a temperature between 34 °C and 36 °C with a period of transformation of 40 minutes. In this phase, the result is a semi-finished cheese, also called "white form".

3.  *Seasoning*: The semi-finished cheese is transported to a particular warehouse (usually not near the dairy), where is left to mature for at least three months. Seasoning must occur in warehouses with the following characteristics: humidity of at least 90%, and temperature between 5 and 12 °C. Consorzio Fontina experts check the entire cheese production after this period and, if it passes the test, the cheese is branded with the exclusive logo of the "Consorzio Tutela Fontina—CTF" (Fontina Protection Consortium).

4.  *Packaging*: When the seasoning phase is complete, the entire cheese is stored and transported at refrigeration temperature between 0° and +4 °C to the last actor in the chain, who takes care of the packaging. The packing and portioning of Fontina cheese are done in the seasoning area to guarantee the preservation of the product characteristics up to the final consumer. The materials used to package the Fontina PDO cheese are a protective atmosphere, polypropylene film, and food paper.

5.  *Sales*: Finally, the whole cheese is divided into portions (0.8–2 kg) and distributed on the market. The shelf life for the entire Fontina PDO cheese is 150 days. While for the portions 63 days.

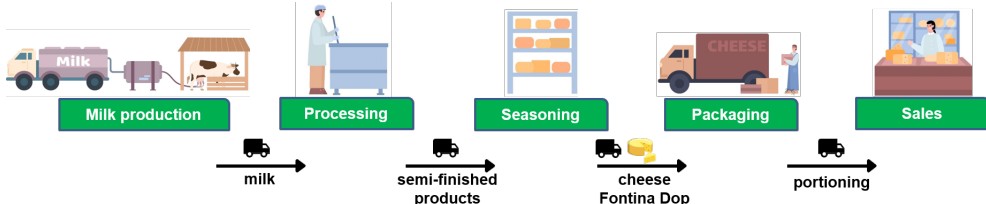

**Figure 2.** The production chain of cheese Fontina PDO.

### 3.2. Fontina PDO—Current Traceability System

Field research was carried out at dairies and mountain pastures. It made it possible to study the current traceability system of Fontina PDO cheese, more focused only on the milk production phase.

The traceability system of Fontina PDO cheese is managed in collaboration with the CTF. The CTF verifies that all the requirements have been met to assign the quality brand. Based on the requirements provided by the consortium and EU regulations, the operator collects information for each phase described before.

The Fontina PDO supply chain already has a system to trace the milk produced by each farmer. The system is employed in the milk production phase. During this phase, the transporter checks the milk conferred by farmers and inserts the quantity (in liters) into the

system. Figure 3, shows the receipt that the system produces when the transporter delivers the total volume of milk collected in mountain pastures to dairies. The systems employed already detect important information for traceability: *Farmer code* for the identification of farmers, *Time and Date* when the milk is collected, the number of *Liters* conferred by each farmer, its *Temperature*, and the total *Count* of delivers.

After this phase, dairy and seasoning operators collect other information manually, writing them every day on a paper form, which causes various potential problems due to the significant amount of information to be traced.

The traceability platform will integrate the existing system to trace information up to the sales phase. Furthermore, the immutability of data will be ensured by the integration of Blockchain technology.

| - Ride start 2021-01-19 Time 05:29:06 am<br>- Tour id: 1<br>- Driver id: 101<br>- Vehicle XN***B | | | | |
|---|---|---|---|---|
| **Farmer Code** | **Time** | **Liters** | **Temp** | **Nr** |
| 000051 | 05:50 | 188.2 | 29.3 | 001 |
| 000128 | 05:50 | 25.85 | 26.0 | 002 |
| 000104 | 06:07 | 208.2 | 26.2 | 003 |
| 000081 | 06:11 | 17.20 | 22.3 | 004 |
| 000042 | 06:26 | 295.0 | 29.5 | 005 |
| 000125 | 06:43 | 46.70 | 27.6 | 006 |
| 000038 | 06:50 | 181.7 | 30.1 | 007 |
| 000126 | 06:55 | 248.8 | 26.3 | 008 |
| - Ride end 2021-01-19 Time 07:02:13 am<br>- Total receptions: 008<br>- Total milk volume: 1211.65 (liters) | | | | |

**Figure 3.** Milk collecting system, Fontina PDO cheese supply chain.

## 4. Results: Fontina PDO Cheese Traceability Platform Based on Green Blockchain

The traceability system based on Green Blockchain was developed and implemented based on the questionnaires and focus group carried out. The following paragraphs show the architecture, the type of information that operators insert into the system, and the developed platform's testing and validation.

### 4.1. Implementation Platform

The traceability platform employed is based on a web application usable by all operators (i.e., farmers, dairies, seasoning workers, transporters and supervisor authorities.) involved in the Fontina PDO supply chain. It allow operators to record information from the dairy supply chain to guarantee the quality and promotion of Fontina PDO cheese. The consumer will be able to consult the transactions in the different supply chain stages. Figure 4 shows the high level of the architecture of the Fontina PDO production chain. The main components of the architecture are:

- *Back-end framework;*
- *Relational Database Management System (RDBMS);*
- *Distributed ledger technology (DLT).*

The traceability platform is developed based on functional and technological requirements collected in the dairies in Aosta valley. The back-end framework represents the software part of the system and contains all the fundamental functions for entering and reading data. The authors employed the Django full-stack [6,7] framework based on Python programming language and with an open-source license. The authors identified

the mandatory data and necessary tables and fields to instantiate in the RDBMS. The database employed is MySQL, which is hosted on Amazon Web Services (AWS). The DLT integrated with RDBMS stores the key fields for identifying transactions from the relational database, allowing the immutability of the data of the Fontina supply chain. Based on the analysis and evaluation of various Blockchain Technologies, the authors choosed to use the Algorand Blockchain, one of the greenest Blockchain based on low power consumptions consensus algorithm.

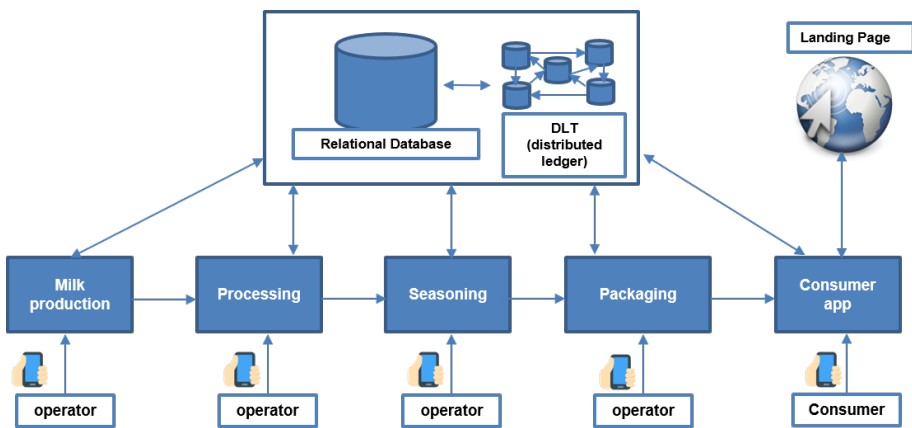

**Figure 4.** Traceability platform high-level architecture.

*4.2. Data Model*

A data model was designed based on the Fontina PDO cheese production chain, considering all the stages along the supply chain and operators involved. The data model specifies the tables and fields to be populated in the RDBMS & DLT with certain values. Based on the data model designed, Figure 5 shows the main information registered by the supply chain operators. The information about the operator name and timestamp of the transaction are recorded automatically by the system. Therefore, all operators registers the following information at each phase:

- *Transporter*: The transporter starts the tour of the milk producers and records each amount of milk supplied (in liters) with the preexisting system. The transporter reaches the dairies and logins into the traceability platform and registers: the total quantity of milk collected, the farm kind (mountain pasture or valley), the date-time of raw milk collected, and finally upload the reception image with all the quantities supplied by the farmers.

- *Dairies operator*: The transporter delivers the milk collected to a dairy operator that processes it. In this phase, the operator checks the last milk production transactions and chooses the *milk_production_id* to be processed. Furthermore, they specify the kind of product type (Fontina cheese, or other subproduct), the number of semifinished cheese produced, and the bath number.

- *Seasoning operator*: The semifinished cheese is transferred to a special place to mature for about three months. In this phase, the seasoning operator first selects the *processing_id* that contains the number of semifinished cheese that should mature. Furthermore, they register information about the times of start and end of seasoning. Finally, they register the number of cheese that is branded with CTF.

- *Packaging operator*: The distribution operator first selects the *seasoning_id* and registers the date of packing and shelf life. The operator submits the transaction and obtains the QrCode. In particular, this phase is very important because the system queries all data connected to this phase and stores this data in the Algorand Blockchain.

- *Sales operator/Consumer*: In this phase, the information about the shop's arrival time and the DateTime of QrCode scanning helps to understand in which place the Fontina cheese is purchased and show the product's history on the landing page of Fontina cheese official site to the final consumer.

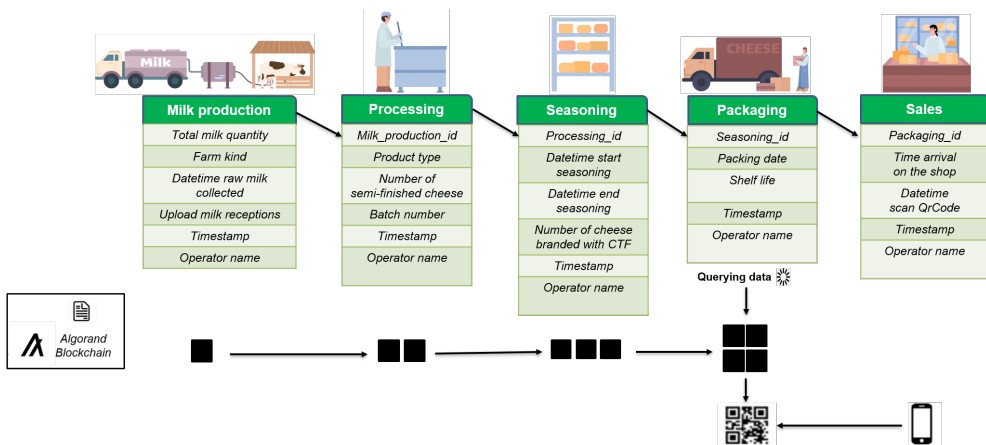

**Figure 5.** Main Information registered by operators.

Each phase of the Fontina PDO supply chain is reported in Figure 5 and contains the name of the table and the id. Every registered table generates an id. All the phases are linked to the id of the generated table, allowing a complete traceback of the system.

The data recorded up to the packaging stage are stored in MySQL and in the Algorand Blockchain. The data are sent to the blockchain once the packaging phase is completed. The system uses a query and traces all the phases connected to the last recorded transaction.

*4.3. Testing & Validation*

The tests and validation of the developed platform were carried out in the dairies located in Aosta Valley (Italy). Before carrying out the tests, digital wallets were created with Algorand Blockchain coins (also called "Algo") for supply chain operators.

Figure 6 shows some screenshots of the developed traceability platform. In particular, the first two screenshots show the interfaces (in Italian) for login in the application and the information to be registered in the milk production phase. While the third and fourth screenshots show the querying data when the packaging phase is complete and the Algorand wallet [54] connected to the traceability platform. In particular, the data is encapsulated in JSON format to be sent to Algorand Blockchain as advanced transactions. Each operator has its own wallet on Algorand, and the JSON generated is sent to a node in a Blockchain to guarantee the immutability of the data registered.

More specifically, once the packaging phase is completed and the transaction is registered on the Blockchain, the traceability system generates a QR Code with the relative link, which allows checking the transaction registered on the Algorand portal ("Algo Explorer"). An example of a registered block with JSON data of the Fontina PDO production chain is available online [55].

However, the authors choose to query the data in the packaging phase to have no impact on costs on the supply chain and be more efficient in terms of electrical power consumption.

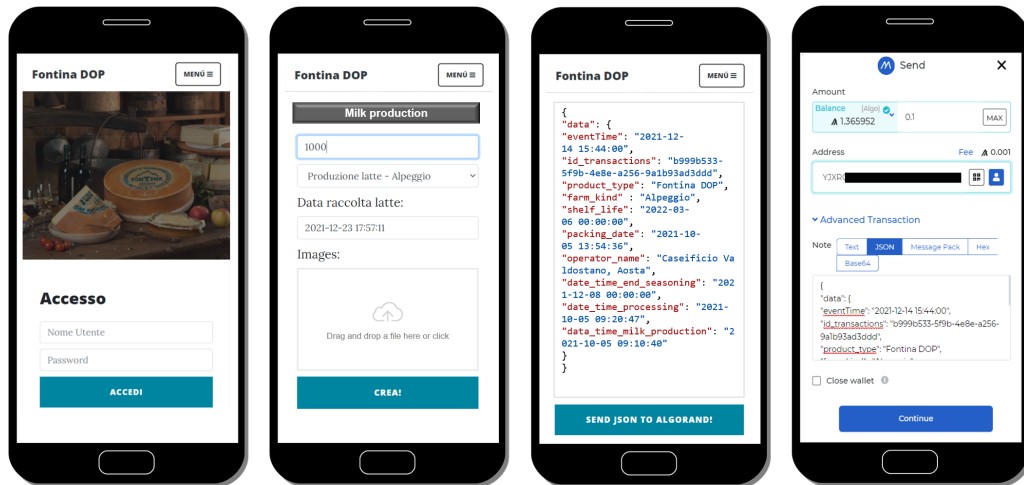

**Figure 6.** Graphical User Interface & the JSON generated sent to the Algorand Blockchain.

The research project brought many benefits to the consortium members. The intervention on almost all supply chain stages permitted the digitization of much of the information and made it immutable. The consumer will be able to consult the information and transactions recorded on the traceability platform integrated with Blockchain Technology directly by scanning the QrCode.

## 5. Discussions

The present article showed an implementation of a Blockchain-based traceability platform applied to the Fontina PDO cheese supply chain.

The architecture and data model has been designed to link the database with the Algorand Blockchain to register transactions and guarantee data immutability. The operators of the Fontina PDO cheese supply chain record all the transactions but send the JSON data to the Algorand platform only at the packaging phase [55]. Due to these choices, the platform has low energy consumption, minimum environmental impact, and overall cost savings. The developed platform has been tested and validated in the dairies in Aosta Valley (Italy).

Many other studies have proposed conceptual frameworks [13–15,37,44], proof of concept [8,47], and data models [45,51] to integrate traceability solutions with emergent digital technologies in the agrifood industry. Some of these have implemented traceability platforms based on IoT and centralized databases [46,48], other studies have implemented traceability platforms integrated with Blockchain with traditional consensus mechanisms [12,49].

In this work the developed Blockchain-based traceability platform has several innovative aspects as compared to other studies that are summarized below:

*All production chain phases digitalized*: The developed platform allows all operators to record information in all supply chain stages. As a result, this increases the information exchange between consortium members.

*Immutability of data collected*: Blockchain guarantees that the data is not changed, improves the production chain's transparency, and valorizes data for the consumer.

*Environmental sustainability*: In terms of environmental sustainability, it was decided to use a Green Blockchain with a low energy consumption consensus mechanism. With Pure Proof of stake algorithm consensus, the transactions are validated with very low energy impact and in a few seconds.

*Cost*: The cost of transactions to be sent to Blockchain is another important aspect. The traceability platform can send a single transaction per every registered phase. Therefore, the costs of issuing transactions into the Blockchain may change. As of February 2022, the transaction fee for the Algorand Blockchain is 0.001 Algo (less than 0.01 cent USD) per

transaction. This amount is based on the exchange rates of Algo/USD. Finally, the authors have chosen to query the data once the packaging phase is complete to avoid high costs that can impact the production chain.

Some of these results coincide with the findings in the relevant literature on the application of Blockchain technology to the dairy milk supply chain [12,49]. However, concerning the use of Blockchain Technology, it is essential to note that no one of the studies has mentioned in their findings the possibility of using a different validation mechanism of transactions with low energy consumption and cost savings.

Moreover, as have mentioned before the cost aspect is very important in small production chains like Fontina PDO cheese. Reducing the number of transactions by collecting all the data along the supply chain and sending them only when all the phases are completed is fundamental to no impact with transaction costs of Blockchain Technologies. For this, the proposed platform is suitable in particular for small local productions such as Fontina PDO, in which all the raw materials are produced in the Aosta Valley region and where producers have a certain level of trust.

The proposed results offer a new perspective on the use of Blockchain Technologies. Above all, in light of the energy crises in progress [56], it is essential to use the technology sustainably [57]. Therefore, third generation Blockchains with low environmental impact must be considered in subsequent studies.

However, the most important limitation of this work is that many operators are not digital-oriented. The use of a digital wallet to send a transaction to the Blockchain platform can be complicated.

## 6. Conclusions and Future Development

The present article proposes a new traceability platform based on the Algorand Blockchain to trace the supply chain of Fontina PDO cheese with minimum environmental and cost impact. Considering the substantial similarity between the existing cheese production chains, the proposed system architecture, design, and interactions inside the entire supply chain can be applied to any other dairy product chain.

Thanks to the Blockchain-based traceability platform developed, the information collected along the Fontina PDO supply chain is available in real-time, improving data exchange & transparency between consortium operators. Finally, the Blockchain guarantees that data is not changing.

The Green Blockchain employed with a lower power consumption consensus mechanism is an excellent value for the operators and consumers in terms of sustainability and the "Made in Italy" promotion. In addition, reducing the number of transactions by collecting all the data along the supply chain and sending them only when all the phases are completed is fundamental to no impact with transaction costs of Blockchain Technologies.

Future developments of the work may include using a fully decentralized application (DAPPS) integrated with smart contracts.

**Author Contributions:** Conceptualization, G.V., G.C., and F.B.; methodology, G.V. and G.C.; software, G.V. and F.B.; validation, L.V.-P., F.B. and G.C.; formal analysis, G.V. and F.B.; investigation, G.V. and G.C.; resources, L.V.-P.; writing—original draft preparation, G.V., F.B.; writing—review and editing, G.V., F.B., G.C., L.V.-P.; supervision, G.C.; project administration, G.C. and O.T.; funding acquisition, G.C. and O.T.; All authors have read and agreed to the published version of the manuscript.

**Funding:** This research was supported by the project: TYPICALP—"TYPicity, Innovation, competitiveness in Alpine dairy Products". Interreg European Project ID: 493717. interreg-italiasvizzera.eu/progetti/typicalp (accessed on 15 Febraury 2022).

**Institutional Review Board Statement:** Not applicable.

**Informed Consent Statement:** Not applicable.

**Data Availability Statement:** Not applicable.

**Conflicts of Interest:** The authors declare no conflict of interest.

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
