# Peer review of "Traceability Platform Based on Green Blockchain: An Application Case Study in Dairy Supply Chain"

_sustainability, doi:10.3390/su14063321_

Round 1
Reviewer 1 Report
1) Please indicate more precisely the aim of the present scientific work in the abstract and at the end of the introduction.
2) In "Methods" please provide what kind of interviews were organised: was there any questionaries’ or it was as a talk? What kind of questions was provided?
3) In "Methods" please provide what is approximate shelf-life of produced cheese (because it is very important during transportation and distributing) and necessary temperature during transportation (because this factor will influence the time of transportation) and will be taken in to consideration during in research system development.
4) please add a scientific discussion to the results section compared to the findings in the scientific literature, there is currently no discussion.
5) Please shorten the conclusions a little more precisely, based on the results of the research.
Author Response
"Please see the attachment."

Reviewer 2 Report
Abstract
The objective of study should be specifically mentioned in the abstract. Refer (Line 64). Mentioning the new traceability platform for the Fontina DOP cheese supply chain is insufficient.
Literature Review
Cost saving is one of the important innovations in the study for the proposed platform of the dairy supply chain. It is importantly included in the study conclusion (Line 370) but it was not specifically mentioned in the study objective (Line 64). Cost saving is a direct result of low energy consumption (Line 136) due to the adoption of the Algorand.
Please include literature on the sustainability aspect which related to the objective of the manuscript.
Other errors
The sentences in lines 227-228, 297, 323-326, 367-269 need rephrasing to convey its accurate meaning.
There are grammatical errors that concern the use of proper tenses (Line 151, 160, 162, 164, 204, 297).
Author Response
"Please see the attachment."

Reviewer 3 Report
I have carefully reviewed the manuscript "Traceability Platform based on Green Blockchain: An Application Case Study in Dairy Supply Chain."
In my opinion, the paper should have revisions before being published.
- The DOP acronym is in Italian, and this is not allowed. Should be replaced by PDO one in lines 5, 31, 61, 66, 70, 224, 232, 237, 238, 259, 261, 281, 283.
- Add references in the paragraph in lines 14-23.
- The paragraph dealing with the dairy supply chain (lines 24-36) does not appear to be in the appropriate position. The text was not smooth. The authors begin the introduction with traceability systems, then go to diary production, and then return to the traceability system. What a mess!
- Avoid the use of the first person in lines 65-66.
- The MAIN problem of the paper was the conciseness in describing the results. Only 73 lines (on 377 ones) have been used in the article to explain the results, the same lines of the introduction (68 lines) and much less than the literature review section (150 lines).
Author Response
"Please see the attachment."

Reviewer 4 Report
The authors submit a relatively brief study for assessment. Anyhow, this is a study whose scope corresponds to the scope of conference papers. For publication in a renowned journal, it would be necessary to revise the article ( in apart from the appropriately chosen standardize its structure), with significant scale and content adjustments. Among other things, justification of the study in relation to the researched topic, appropriate choice of research question based on the defined problem, describe in more detail the methods and their justification in relation to the identified research problem, etc. In my opinion, the identified shortcomings go beyond possible adjustments. The topic is current and relevant, its deeper research will certainly bring the authors the expected (publication) results.
Author Response
"Please see the attachment."

Round 2
Reviewer 1 Report
1) In methods please mention the storage temperature of cheese (it was not mentioned) and precise the name of packaging material, not only cellophane.
2) In methods you mention: Temperature “Processing: The transporter delivers the total milk collected to dairies for transformation.
The milk delivered must not have undergone heating to a temperature above 36 °C.” But the question was about storage temperature of the cheese.
3) My suggestion was: please add a scientific discussion to the results section compared to the findings in the scientific literature, there is currently no discussion.
In added answer not references was mentioned. Under scientific discussion I mean you obtained data comparison with data from scientific literature. At the moment please provide a little bit dipper discussion with more comparison with scientific literature.
3) Conclusions are not same as a summary. Please make it shorter with some data.
Author Response
"Please see the attachment."

Reviewer 3 Report
All my queries have been complied.
Author Response
Dear Reviewer Thank you very much for your response!
We are glad that our revisions satisfied your comments.
Best Regards!
Reviewer 4 Report
The authors (at first glance) have improved the quality of their study. In any case, it is difficult to orient oneself in the presented document, the changes in the text are not indicated in any way. In terms of form, the text presented is in principle compatible with the works published in the journal. However, in terms of content, my original comments and concerns remain. I do not recommend publishing the study in the chosen journal.
Author Response
Dear Reviewer,
Thank you very much for your comment.
We hope that our new revisions have improved some of the most critical points cited by you.
Best Regards
Round 3
Reviewer 1 Report
Conclusions:
please remove any data from literature, because in the conclusions only original author data will be summarised, no literature resources.
Author Response
Dear Reviewer, we want to thank you for your very helpful and constructive feedback.
Response to Reviewer 1 Comments
1) Conclusions:
please remove any data from literature, because in the conclusions only original author data will be summarised, no literature resources.
Response
Thanks for the suggestion. We removed the literature resources and updated the paragraph related to the Conclusion section:
The present article proposes a new traceability platform based on the Algorand Blockchain to trace the supply chain of Fontina PDO cheese with minimum environmental and cost impact. Considering the substantial similarity between the existing cheese production chains, the proposed system architecture, design, and interactions inside the entire supply chain can be applied to any other dairy product chain.
Thanks to the Blockchain-based traceability platform developed, the information collected along the Fontina PDO supply chain is available in real-time, improving data exchange & transparency between consortium operators. Finally, the Blockchain guarantees that data is not changing.
The Green Blockchain employed with a lower power consumption consensus mechanism is an excellent value for the operators and consumers in terms of sustainability and the "Made in Italy" promotion. In addition, reducing the number of transactions by collecting all the data along the supply chain and sending them only when all the phases are completed is fundamental to no impact with transaction costs of Blockchain Technologies.